# How *Porphyromonas gingivalis* Navigate the Map: The Effect of Surface Topography on the Adhesion of *Porphyromonas gingivalis* on Biomaterials

**DOI:** 10.3390/ma15144988

**Published:** 2022-07-18

**Authors:** Retno Ardhani, Rasda Diana, Bidhari Pidhatika

**Affiliations:** 1Department of Dental Biomedical Science, Faculty of Dentistry, Universitas Gadjah Mada, Yogyakarta 55281, Indonesia; retnoardhani@mail.ugm.ac.id; 2Audy Dental Clinic, Jakarta 17214, Indonesia; 3Research Center for Polymer Technology, National Research and Innovation Agency, Republic of Indonesia—PRTPL BRIN Indonesia, Serpong, Tangerang Selatan 15314, Indonesia; bidhari.pidhatika@brin.go.id

**Keywords:** *Porphyromonas gingivalis*, surface topography, dental implants, subperiosteal implant, surface roughness, depth profile

## Abstract

The main purpose of this study is to develop an understanding of how *Porphyromonas gingivalis* responds to subperiosteal implant surface topography. A literature review was drawn from various electronic databases from 2000 to 2021. The two main keywords used were “*Porphyromonas gingivalis*” and “Surface Topography”. We excluded all reviews and or meta-analysis articles, articles not published in English, and articles with no surface characterization process or average surface roughness (R_a_) value. A total of 26 selected publications were then included in this study. All research included showed the effect of topography on *Porphyromonas gingivalis* to various degrees. It was found that topography features such as size and shape affected *Porphyromonas gingivalis* adhesion to subperiosteal implant materials. In general, a smaller R_a_ value reduces *Porphyromonas gingivalis* regardless of the type of materials, with a threshold of 0.3 µm for titanium.

## 1. Introduction

Subperiosteal implants were first introduced in Sweden in 1942 as an alternative to treat patients with severely atrophic bones [1]. These implants were designed to be placed in between the bone and the periosteum. The main idea was to distribute stress from the prostheses to a large area of bone support [2]. Although, in recent years, subperiosteal implants have been gradually replaced by endosseous implants, for the patient with severely atrophic bones, this type of implant is irreplaceable [3]. In addition, with advanced technology such as computed tomography (CBCT), intraoral scanners, computer-assisted-design/computer-assisted-manufacturing (CAD/CAM) software, and newly discovered materials, the subperiosteal implant has started to regain its popularity [4]. However, the subperiosteal implant has several disadvantages, such as a complex fabrication technique, time-consuming procedures, and a higher risk of postoperative complications [2,5,6]. One of the most common postoperative complications is an infection on the permucosal abutment post. This infection has a clinical characteristic similar to peri-implantitis on the endosseous implant. Once the infection spreads, the option is to perform tissue resection or complete the implant removal [7].

Peri-implantitis is an inflammation around implants and induced progressive bone loss [8]. The effect of peri-implantitis is generally to cause more significant bone loss and more rapid progress than periodontitis [9]. It was found that microorganisms play a vital role in peri-implantitis through biofilm formation [10]. After the implant surface is exposed to the oral environment, the biofilm starts to form in the peri-implant pocket [11]. The biofilm is formed through five stages: (1) reversible cell attachment; (2) irreversible cell attachment facilitated by extracellular polymeric substance (EPS); (3) cells attached on surfaces replicate and form microcolonies; (4) biofilm maturation by forming a three-dimensional structure; (5) detachment of some cells from biofilm and dispersal to propagate and produce biofilm renewal [12]. Overall, biofilm formation happens within 1–2 weeks and reaches its stability after three months [11]. In addition, bacterial infections lead to inflammation, and implant failure can occur at any time during treatment [13,14,15].

To date, to improve the cells and tissue attachment, the implant surface has been modified by both chemical and or physical alteration, which includes creating grooves and roughness [16]. However, implant surface characteristics are crucial not only for tissue attachment but also for biofilm formation. In vitro and in vivo studies reveal that implant surface properties regulate bacterial attachment, physiology, and biofilm formation [12,17]. This is because bacteria can sense chemical signaling and surface-associated mechanical cues. The first clue regarding this phenomenon came in 1981 when Beachy found that different bacteria in the same niche do not interact with the same surfaces. *Streptococcus salivarius*, for example: *S*. *salivarius* binds to the tongue but not to the teeth, whereas *Streptococcus mutans* acts reversely [18]. Similarly, with *P. gingivalis*, within the same salivary pellicle, the addition of peptide base coating inhibits the attachment of these bacteria. On the contrary, a non-coated implant disk showed a higher number of *P. gingivalis* attached [19].

The effect of topography on bacterial adhesion is like two sides of the same coin, which depends on the size, patterns, and distribution of the topography. For example, some studies suggest that implants with micro-roughness have higher biofilm and bacterial accumulation than more refined surfaces [17,20,21]. At the same time, other studies found the possibility of an antifouling effect from micro-topography by changing the surface wettability [22]. Similar to microtopography, some nanotopography also affects bacterial activity on implant surfaces. Studies showed that nanotopography could induce bacteria to produce different types of EPS [23]. Nanotopography also affects bacterial membranes. Nanopillars, for example, act like “a bed of nails”, which ruptures bacterial membranes once it is in contact with this surface [22].

Like the microflora in tooth sulci, anaerobic bacteria such as *Staphylococcus aureus*, *Prevotella*, *Porphyromonas gingivalis*, *Bacteroid fragilis*, and *Fusobacterium* are also associated with periimplantitis [17,24,25]. Based on a systematic review, it was found that *Porphyromonas gingivalis* (*P. gingivalis*) was frequently found at the peri-implantitis site [17,26]. *P. gingivalis* is a Gram-negative, obligately anaerobic, non-motile, and non-spore-forming bacterium with several virulence factors: hyaluronidase and chondroitin sulfatase enzymes, lipopolysaccharide (LPS) capsule, fimbriae, collagenase, and aminopeptidase [27]. These virulence factors enable *P. gingivalis* to invade the periodontal tissue surrounding the implants locally. In addition, *P. gingivalis* can cause not only local inflammation on the implant site but also systemic disease via four stages: (1) bacteremia, (2) activation of persistent inflammatory cascades, (3) spread of specific toxins, and (4) pathogens trafficking by direct infection and internalization in host immune cells throughout the body [28]. Thus, this study systematically reviews the data for the effect of micro- and nano-topography on the bacterial activity of *P. gingivalis*, aiming to provide a better understanding in designing subperiosteal implant surfaces to reduce peri-implantitis.

## 2. Materials and Methods

A depth literature review was performed to answer the research question. Data were collected from several online sources such as Google Scholar, Cochrane Library, Science Direct, Wiley, and PubMed. The data taken were published from 2000 to 2021 with the main keywords as “*Porphyromonas gingivalis*” and “Surface Topography”. A total of 1290 articles were found. However, no report specifically reviewed the effect of various surface topographies regarding subperiosteal implants on the activity of *Porphyromonas gingivalis*. From these findings, we excluded all reviews and or meta-analysis articles, articles not published in English, and articles with no surface characterization process or R_a_ value. Further details on article selection can be seen in Figure 1.

## 3. Discussion

### 3.1. Porphyromonas gingivalis Structure and Characteristic

Principally, bacterial surface components and their extracellular compounds, such as fimbriae or pili, LPS, and EPS, combined with environmental conditions and quorum-sensing signals, are critical for biofilm formation [29]. Below, we will discuss the *P. gingivalis* structure and its significance, especially in biofilm formation.

#### 3.1.1. Fimbriae

Bacteria, as well as *P. gingivalis*, commonly express their extracellular polymer known as pili or fimbriae, which are similar terms [30]. Fimbriae are considered significant factors in determining *P. gingivalis* virulence, as fimbriae help in bacterial adhesion to the host surface, antibiotic surface, and or in between bacterial cells [31,32,33]. In an in vitro polymerase chain reaction assay (PCR), it was found that the number of fimbriae in a *P. gingivalis* strain is equal to its ability to adhere [32]. A *P. gingivalis* mutant 33277 (MPG1) with minimum fimbriae could not adhere to both epithelial cells or the gingival fibroblast [34]. Fimbriae are also classified into several types based on several classification schemes. The most popular one is a classification based on its morphology and function by Brinton in 1965. Brinton classified six types of fimbriae and then, one year later, Duguid et.al. added a seventh (Type 1 to 6 and F) [33]. However, with *P. gingivalis*, the widely used classification is based on the nucleotide sequences, in which six genotypes of the *fimA* gene have been identified (*fimA* I, Ib, II, III, IV, and V) [35].

*P. gingivalis*, in general, has two types of fimbriae, which are *FimA* and *Mfa1* fimbriae. The *FimA* fimbriae are composed of *FimA* proteins encoded by *FimA* genes and called the long fimbriae. Similarly, the *Mfa1* fimbriae are composed of the *Mfa1* protein encoded by *Mfa1* genes, and are called the short fimbriae [30,36,37]. These fimbriae help bind specifically to and trigger various host cells, such as epithelial, endothelial, and spleen cells, as well as peripheral blood monocytes in humans, resulting in the release of several distinct adhesion molecules and inflammatory cytokines [30]. In addition, there are several accessory proteins which are incorporated into fimbriae; for example, *Mfa4* which are incorporated into *Mfa1* fimbriae. *Mfa4* mediate the formation of *Mfa1* by promoting the maturation of *Mfa3* and stabilizing *Mfa5* within the cell surfaces; thus they are crucial in biofilm formation [38].

#### 3.1.2. Capsule

The *P. gingivalis* strain exhibits significant heterogeneity, in which some strains are encapsulated, whereas others are non-encapsulated [39]. Previous studies have reported that the encapsulated strain of *P. gingivalis* has higher virulence than the non-encapsulated one. The capsule plays a major role in evading host immune system activation, reducing phagocytosis, increasing bacterial activity survival ability within the host cells, and boosting its virulence [39,40,41].

#### 3.1.3. Cell-Wall

In the Gram-negative bacteria, such as *P. gingivalis*, the cell wall is formed from a single layer of peptidoglycan covered by a membranous structure called the outer membrane vesicles (OMVs). *P. gingivalis* expresses protease activity which can be extruded with the OMVs [42]. OMVs enable bacteria and host communication as they can carry molecules involved in immune modulation [43]. *P. gingivalis* OMVs are adherent and small, with the ratio of cells to OMVs at approximately 1:2000 [44].

### 3.2. Biofilm Formation

Biofilm is a microbial community attached to the interface enclosed in an EPS that exhibits a distinct phenotype correlated to its gene transcription and growth rate. It is known that the biofilm has been shown to have a specific mechanism for initial attachment to a surface, development, and detachment [45]. Overall, it is believed that biofilm formation begins with bacterial attachment on the surface, which transforms from reversible to irreversible. Adhesive components of bacteria aid this transformation. This attachment then advanced through EPS production, which later entrapped the whole structure. Finally, some bacterial cells escape from the mature biofilm to form new colonies [46]. Once the biofilm is developed, killing the bacteria inside or removing the biofilm from the surface becomes difficult. Bacteria inside the biofilm are packed and resistant to the adverse environment, for example, antibiotics [47]. Hence, interspersing initial bacteria attachment, including their sensing mechanism, is crucial to preventing biofilm formation and related problems [48].

Biofilm formation on the subperiosteal implant is affected by several factors such as (1) oral environment, (2) bacterial properties, and (3) material surface characteristics, including chemical composition, surface free-energy, hydrophilicity, and surface topography (roughness) [17,49]. Higher surface free energy has shown significant correlation to bacterial adherence. Higher surface free energy favours bacterial attachment [50]. In addition, the combination of surface free energy and surface roughness is the major factor and proportional to surface hydrophilicity with low surface energy and smoother-surface-producing higher hydrophobicity [51]. Almost all in vivo studies suggest that a smooth surface reduces the amount of biofilm compared to a rough one. An increase in surface roughness of more than 0.2 μm and or an increase in surface energy promotes biofilm formation, with surface roughness being more dominant [49].

### 3.3. Surface Topography

Physical modification of surfaces can provide long-term effectiveness and is environmentally friendly. Thus, the physical modification is believed to be a more promising alternative compared to the chemical modification of surfaces [52]. One of the important parameters in the identification of physical surface properties is surface topography, which refers to both the profile shape and the surface roughness, including the waviness and the asperity or the finish [53].

Furthermore, the most frequently used parameters for characterizing surface topography are average surface roughness (R_a_) and root-mean-square surface roughness (R_rms_), that stands for the average and root-mean-square deviation of height values from the mean line, respectively. However, both R_a_ and R_rms_ provide no information on the spatial distribution or shape of the surface features. Some researchers have offered new parameters for a more comprehensive characterization of the surface topography, such as summit density (S_ds_) and developed area ratio (S_dr_) [52].

In the next paragraph, the surface roughness is presented in R_a_ or S_a_. The average roughness, R_a_, provides a general measure of the height of the texture across a surface. It is the average of how far each point on the surface deviates in height from the mean height, while S_a_ is an absolute value that expresses the difference in height of each point to the arithmetical mean of the surface [54]. In general, surface energy (often presented as water contact angle) changes as the surface roughness changes [55]. However, one should keep in mind that surface chemistry also plays key roles in the changes in surface energy [56,57,58].

### 3.4. Subperiosteal Implant Materials and Surface Modification

Material selection in subperiosteal implant placements plays a key role in implant success [59]. In general, like the endosseous implant, subperiosteal implant material is divided into three categories which are metal, ceramic, polymer, and composite [60]. To improve materials properties, surface treatment is commonly applied. The addition of surface treatment improves cell attachment and bacterial debridement. Surface treatment is arguably the most studied topic regarding implant design alteration. There are various types of surface treatment; however, they can be simplified into two types which are chemical and physical. Both of these types showed efficacy in increasing bone attachment and or bacterial debridement [61]. In this section, we will discuss the materials used for subperiosteal implants and the various surface modification methods applied.

One of the widely used materials for subperiosteal implants is titanium and its alloys [59,62]. Titanium and its alloys are still a material of choice for dental implants, as they have a high success rate, are durable, and display adequate osseointegration [63]. There are several methods used for titanium surface modification such as sandblasting, acid etching, a combination of both sandblasting and acid etching (SLA), fluoride treatment, calcium phosphate coating, and anodic oxidation [64,65]. Among these methods, sandblasting is one of the most popular. Sandblasting or acid etching or a combination of both can increase surface roughness, increasing the surface area for osteoblast attachments. Hence, it increases bone healing, interfacial stress distribution, and bonding strength [66]. In addition, Alagatu et al. [59], mentioned sandblasting as the best method for titanium and zirconia.

Recently, the popularity of zirconia as an alternative for implant materials has increased. In their review, Alagatu et al. [59] showed that some clinical studies demonstrated that zirconia has better anti-inflammatory properties than titanium. In addition, zirconia is less prone to peri-implantitis than titanium. Zirconia also can be combined with titanium to improve both properties. The addition of zirconia increases implant biocompatibility compared to titanium alone [67]. Several attempts have been made to improve the properties of zirconia such as the addition of hydroxyapatite [68] or calcium phosphate [69], sandblasting [70], acid etching [71], laser treatment [72], and ultraviolet photo-functionalization [73].

### 3.5. How Porphyromonas gingivalis Responds to Topography

An increase in roughness can increase *P. gingivalis* attachment; for instance, with ceramic material. The type of ceramic did not affect *P. gingivalis*’s LPS adherence; however, surface roughness does [74]. Furthermore, Verran & Boyd have classified surface roughness into three categories based on the roughness average (R_a_): macro (R_a_~10 µm), micro (R_a_~1 µm), and nano (R_a_~0.2 µm) [75]. Previous research found that with bacteria with relatively thin cell walls such as *P. gingivalis*, bacterial attachment is mainly affected by roughness [76,77]. Moreover, it has been suggested that nano roughness is appropriate to prevent microbial adherence. It is because most bacteria, as well as *P. gingivalis*, are about 1.51 µm long and 1 µm in diameter [77,78]. In addition, it has been reported that based on numerous in vitro studies in fixed restorations, the degree of bacterial attachment increases with increasing surface roughness greater that 0.2 μm [58]. In this section, we will discuss the effect of various topographies on several subperiosteal implant materials on the activity of *P. gingivalis*.

Using the profilometry method, Zortuk et al. [58] studied the surface roughness of cylindrical substrates based on various bis-acrylic composites, namely Dentalon (R_a_ = 1.41 ± 0.36 µm), Revotek LC (R_a_ = 2.30 ± 0.43 µm), PreVISION CB (R_a_ = 1.82 ± 0.62 µm), Protemp 3 Garant (R_a_ = 1.10 ± 0.49 µm), and glass as the control substrate (R_a_ < 0.01 ± 0.00 µm). Using the spectro-fluorometric method, the authors then investigated how the topography and the chemical nature of the substrates affect the attachment of *P. gingivalis*. The highest and the lowest bacterial attachment were found on the roughest (Revotek LC) and the smoothest (glass) surface, respectively. Interestingly, the bacterial adhesion was greater on Protemp 3 Garant compared to both Dentalon and PreVISION CB. These findings demonstrate that both the surface roughness and the chemical nature of the substrates play roles in *P. gingivalis* attachment on surfaces.

Daw et al. [79] prepared various scales of roughness on a titanium surface by means of hydrogen peroxide treatment from 0 h, 6 h, 24 h, 1 week, to 4 weeks, which resulted in Ra = 0.83 ± 0.08, 1.03 ± 0.18, 1.22 ± 0.12, 2.78 ± 0.44, and 3.14 ± 0.53 µm, respectively. It was found that the moderate- to high-scale surface roughness titanium surface with a R_a_ value of 1.2 µm and 2.7–3.2 µm, respectively, increases *P. gingivalis* attachment in vitro due to the presence of surface depression, edge, and pits. These features enhance contact between *P. gingivalis* and the surface and protect *P. gingivalis* from hydrodynamic shear forces [79]. Another article by Han et al. [64] reported various surface roughnesses on zirconia ranging from R_a_ = 0.17 ± 0.03 µm (untreated), 0.56 ± 0.05 µm (grit-blasting), 1.47 ± 0.04 µm (HF-etching), to 1.48 ± 0.05 µm (grit-blasting followed by HF-etching). Furthermore, the roughness data were complemented with water contact angle (WCA) data that indicate the surface energy. It was shown that the WCA value decreases with increasing surface roughness, meaning that the surface energy at a solid-water interphase decreases (more hydrophilic) with increasing surface roughness. The bacterial assay showed that within the first 24 h, the surface with the highest R_a_ (the last two treated surfaces) exhibit the highest *P. gingivalis* attachment. Interestingly, the corresponding surfaces exhibit the lowest *P. gingivalis* biofilm after 3 × 24 h. The authors speculated that the low biofilm accumulation after 3 days correlates to the low surface energy and hydrophilicity (indicated by the low WCA). In agreement with previous studies, the authors then concluded that after the maturation of the biofilm, the influence of surface roughness on bacterial attachment diminishes [64].

Kim et al. [80] treated titanium disks with mechanical grinding, grit-blasting, sandblasting, and Mg ion implantation. The mentioned treatments resulted in 4 (four) different surfaces, namely ground/control (G), sandblasted (S), ground with Mg ions (Mg-G), and sandblasted with Mg ions (Mg-S) surfaces. SEM and profilometry data showed significant increases in surface roughness when sandblasting was applied to the ground sample, that is from R_a_ = 0.61 ± 0.03 to R_a_ = 1.14 ± 0.19. The in vitro experimental results showed a clear, positive relationship between surface roughness and *P. gingivalis* attachment; that is, significantly more bacteria attached to the rougher surface compared to the smoother one. Interestingly, implantation of Mg ions did not alter the surface roughness of each surface; however, it significantly altered the *P. gingivalis* attachment. These phenomena demonstrate that roughness does play a role; however, it is not the only factor affecting *P. gingivalis* attachment to surfaces.

Moreover, Batsukh et al. [81] showed that the surface debridement method using ultrasonic scaler resulted in a rougher surface with R_a_ = 0.5174 ± 0.12 µm compared to the rubber-polished, Ga-Al-As-lasered, and chlorhexidine-treated surfaces with R_a_ = 0.1772 ± 0.04, 0.2119 ± 0.02, and 0.2028 ± 0.01 µm respectively. The highest *P. gingivalis* attachment was found on the roughest surface (highest R_a_). In line with this result, the lowest growth of biofilm was found on the smoothest surface. An interesting finding was reported by Mukaddam et al. [82], in which a titanium surface with nanospikes of 0.5 µm in height prepared from helium sputtering exhibited lower *P. gingivalis* attachment compared to smooth-machined and sandblasted and acid-etched titanium surfaces. It was found that the nanospikes of 0.5 µm in height induced dysmorphisms within *P. gingivalis* cultures following the incubation period because when attached to the surface, *P. gingivalis* appears to be stretched or deflated [82]. Another study reported that both titanium and zirconia with similar R_a_ of 0.21 ± 0.06 and 0.22 ± 0.03 μm, respectively, exhibit similar resistance to *P. gingivalis* compared to a bovine enamel surface with R_a_ = 0.05 and 0.1 μm. Although the bovine enamel surface is significantly smoother, it attracts more *P. gingivalis* due to the charges present on the surfaces, which have boosted the electrostatic interaction with *P. gingivalis*’s cells [83].

A series of in vivo studies have reported that surfaces with roughness above the R_a_ threshold of 0.2 µm are prone to bacterial attachment and biofilm formation [84]. This is because rougher surfaces provide both more attachment sites and “shelter” to the bacteria against shear forces [84]. Moreover, smoother surfaces tend to have higher surface energy at the solid-water interphase (more hydrophobic) [85]. Low surface energy at the solid-water interphase hinders bacterial attachment [56]. This view is supported by Bermejo et al. [17], who reported that the attachment of *P. gingivalis* was significantly higher on a titanium surface with Sa = 1.1–2.0 µm compared to that with Sa = 0.5–1.0 µm.

It was found that laser treatment on titanium Grade 4 effectively reduces *P. gingivalis* biofilm formation. Laser treatment causes a unique surface topography, which appears as craters surrounded by relatively rougher areas (Figure 2). The diameter of the craters is approximately 0.10 µm with a depth of about 0.07 µm [86]. Another article by Xu et al. [76] reported the synthesis of 3 (three) different titanium dioxide surfaces by means of electrochemical anodization, that resulted in nanotubes with 0.03, 0.10, and 0.20 µm-diameter, respectively. Based on atomic force microscopy (AFM) experiments, the surface with 0.1 µm-diameter nanotubes exhibited the lowest roughness (S_a_ < 0.5 µm) compared to the two other studied surfaces. The application of a suitable range of voltage was believed to result in highly ordered and smooth titania nanotubes (TNT) [76]. Furthermore, plate-counting and SEM methods showed a reduced attachment and a round morphology, respectively, of the *P. gingivalis* on the smoothest TNT surface [76]. In contrary, a titanium oxide surface without any treatment presented the greatest roughness (S_a_ > 1.3 µm) and the highest *P. gingivalis* attachment.

Specific nanotopography and a hierarchical arrangement of micro- or nanopatterns on titanium surfaces are proven to alter *P. gingivalis* attachment. A configuration of 100–50–20–10–5 μm width grooves arranged in a parallel direction at 2 μm depth effectively reduces *P. gingivalis* adhesion. It was found that the diameter and the arrangement of 0.055 μm nanotube topography changes the surface contact angle from 87–100° to 35–50° after enlargement of the groove depth from 2 μm to 3.6 μm. A lower contact angle increases surface hydrophilicity [86]. Furthermore, titanium surfaces with different compositions and similar R_a_ of 0.029 µm also showed bactericidal activity against *P. gingivalis* [87].

On the contrary, nanocavities formed by etching using a H_2_SO_4_/30% H_2_O_2_ mixture resulted in a diameter from 0.01 to 0.02 µm and failed to show an antibacterial effect against *P. gingivalis* [78]. Corrosion also causes a R_a_ increase in pure titanium or titanium alloy. Valentim et.al (2014) showed that although pH 3.0 corrosion resulted in the highest R_a_ compared to the rest of the group for both pure titanium (cp-Ti) and titanium-aluminum-vanadium alloy (Ti-6AI-4V) (Figure 3), there was no significant increase in *P. gingivalis* attachment [88]. In addition, a similar titanium R_a_ of about 0.0023 µm showed no differences in *P. gingivalis* growth. Furthermore, various titanium surface modifications with R_a_ of 0.025 and 0.035 µm also did not show significant bactericidal activity against *P. gingivalis*, with various modifications such as oxide nitride or hydroxyapatite [89]. However, the key factor in determining *P. gingivalis* adhesion within those R_a_ is the surface chemistry [89,90]. These results support a previous study which showed that a R_a_ value below 0.2 µm resulted in no further decline in bacterial adhesion [91].

This also accords with ceramics material, which showed that zirconia toughened alumina (ZTA) with R_a_ of 0.031 ± 0.10 µm has lesser *P. gingivalis* attached compared to the ZTA after sandblasting with a R_a_ of 0.465 ± 0.06 µm [92].

## 4. Conclusions

We reviewed the influence of the surface topography, from a configuration and size perspective, on the attachment of *P. gingivalis* on subperiosteal implant materials regardless of the type of materials. Methods to obtain various surface roughnesses included both physical and chemical treatments, such as mechanical grinding, grit blasting, sandblasting, ultrasonic scaling, electrochemical anodization, plasma implantation, hydrogen peroxide treatment, hydrofluoric acid treatment, and so on. The parameters used as the quantification of surface roughness were R_a_ and S_a_, obtained from profilometer and/or AFM. Methods to obtain bacterial attachment data include optical microscopy (fluorescence microscopy) and electron microscopy (SEM). The latter was also used by many authors to acquire the surface topography images. In general, the rougher the surface (higher R_a_ or S_a_), the higher the bacterial attachment. Some studies show, however, that the mature biofilm is no longer affected by surface roughness, but rather by surface energy. Some authors highlighted the fact that both surface roughness and surface energy affect *P. gingivalis* attachment on surfaces, but the influence from surface roughness is greater compared to the surface energy, especially before the maturation of the biofilm. It is also important to highlight the fact that surface roughness is not the only factor affecting *P. gingivalis* attachment on the surface. The inherent chemistry of the substrates also affects the attachment due to variation in polarity, charges, steric repulsion, and so on.

The configuration and size of the surface topography affect the *P. gingivalis* attachment on subperiosteal implant materials regardless of the type of materials. Nano-topography with a size below 0.2 µm is more likely to have more bactericidal effects on *P. gingivalis* than a rougher surface. However, in titanium, the R_a_ below 0.03 µm causes no further reduction in *P. gingivalis*. Due to limited information on other parameters for surface topographical characterization in cell adhesion, this research only includes the R_a_ value. Further research should include other parameters such as summit density (S_ds_), developed area ratio (S_dr_), and root-mean-square surface roughness (R_ms_). These parameters combined might explain the shape and spatial distribution of surface features. Furthermore, to improve subperiosteal implant survival, it is advisable to design a surface treatment in accordance not only with the cell attachment but also to pay attention to its effect on the bacterial attachment.

## Figures and Tables

**Figure 1 materials-15-04988-f001:**
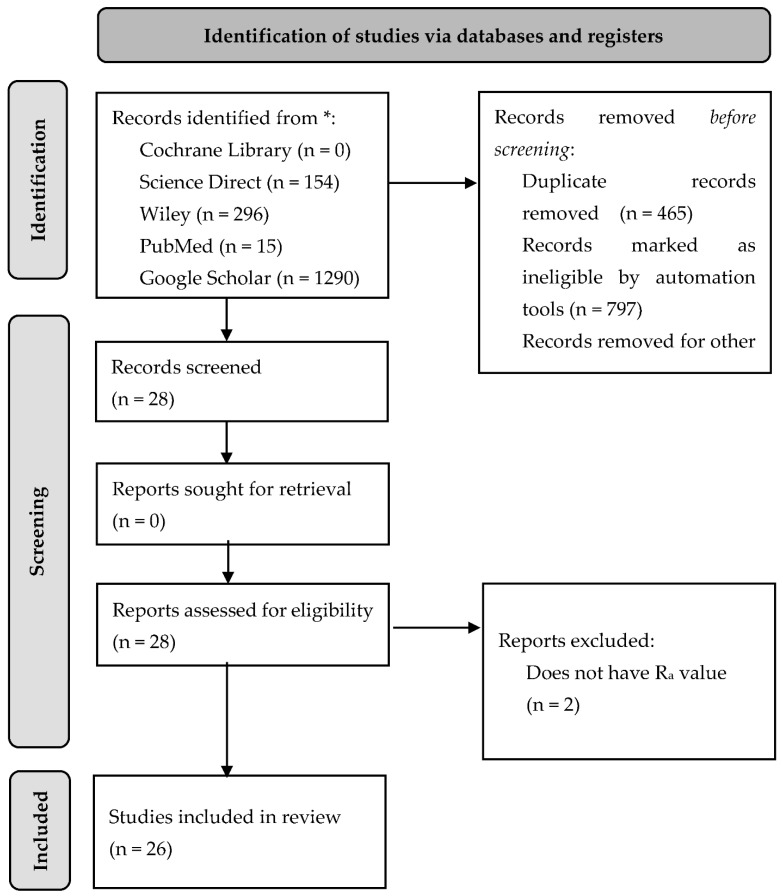
Preferred Reporting Items for Systematic Reviews and Meta-Analysis (PRISMA) based article selection flow chart. Asterisk indicates The Preferred Reporting Items for Systematic Reviews and Meta-Analysis (PRISMA) is a 27-item checklist used to improve transparency in systematic reviews.

**Figure 2 materials-15-04988-f002:**
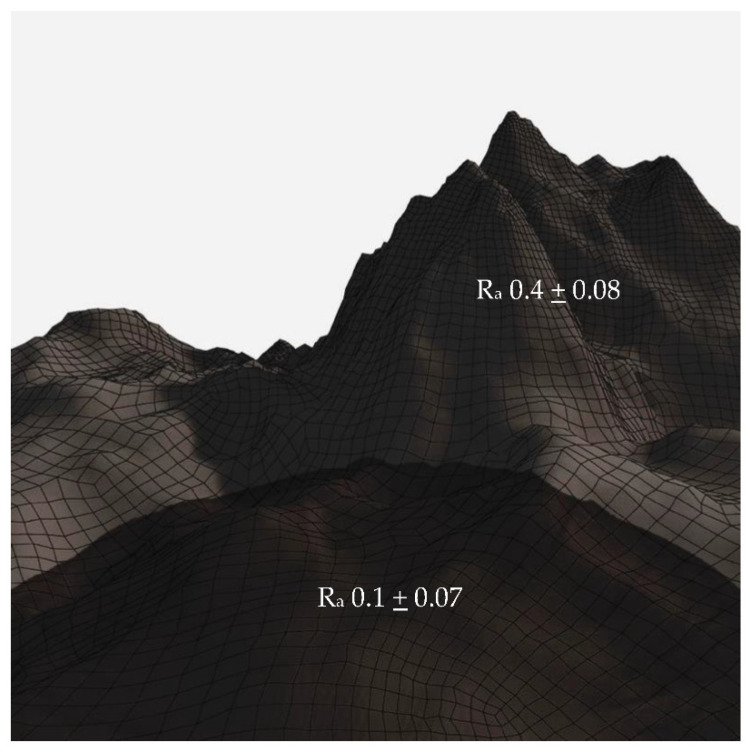
Crater-like topography surrounded with nanospike [86].

**Figure 3 materials-15-04988-f003:**
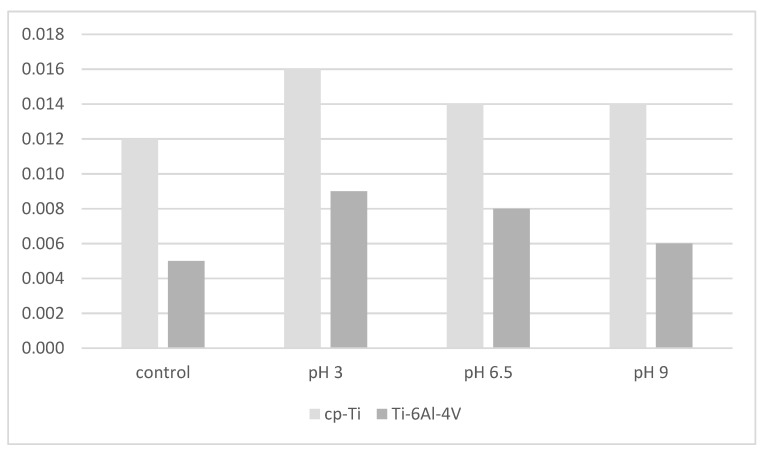
Six-hour attachment of *P. gingivalis* to cp-Ti and Ti-6Al-4V alloys as a function of corrosion at different pHs [88].

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
