# Peer review of "How Porphyromonas gingivalis Navigate the Map: The Effect of Surface Topography on the Adhesion of Porphyromonas gingivalis on Biomaterials"

_materials, 2022, doi:10.3390/ma15144988_

Round 1
Reviewer 1 Report
Title:
How Porphyromonas Gingivalis Navigate The Map: The Effect of Surface Topography on Porphyromonas Gingivalis Activity.
Aim:
The main purpose of this study is to develop an understanding of how Porphyromonas gingivalis respond to subperiosteal implant surface topography
Considering the title and the aim of this review paper, I expected the organization of the main body of the manuscript should be according to the surface topography of the different materials related to dental implants. However, the main body of the manuscript was organized according to the structure of the bacteria (Porphyromonas gingivalis). Is unclear the term Porphyromonas gingivalis activity (which means the capability to adhere to implant materials or other activity); there is no description of this activity in the manuscript and there is no description in the whole manuscript of “activity” in the results and discussion.
The keywords: The authors describe the use of two keywords: Porphyromonas gingivalis and surface topography. It looks like an incomplete search according to the title and aim of the manuscript. Perhaps dental implants, surface roughness, and depth profile should be included as keywords.
Methods described: The instruments used to assess surface roughness and depth are unclear, which is an important issue in this review, the method to evaluate these trails should be mentioned. In addition, it is important to include references divided into studies into invitro and in vivo in the body to provide clarity of different studies analyzed in the manuscript.
I believe that according to the title and aim of the study, it is a better approach to describe the paper according to the materials used in dental implants and its association with the adherence of Porphyromonas gingivalis. In this regard, the external treatment of implants used to improve osteointegration and several treatments produce differences in topography; this issue should be addressed in this manuscript.
The references should be revised since there are inconsistencies, the author did not describe the original methods ( reference 35), and the original research was published by (Amano et al.,1999). Distribution of Porphyromonas gingivalis strains with fimA genotypes in periodontitis patients. J Clin Microbiol 37:1426–1430.
Author Response
In general, we add few additional informations based on your suggestions. Further details, please see the attachment

Reviewer 2 Report
The authors reviewed the literature to find the effect of surface topography on P. gingivalis activity. Since the activity is not a defined term, I suggest modifying the title. The article is well written and interesting but needs further improvement before being considered for publication.
- The title needs to be modified to “The effect of surface topography on the adhesion of Porphyromonas gingivalis on biomaterials.”
- Line 14 abstract, make 26 selected publications as is shown in figure 1.
- The abstract describes that the review article mainly aimed to show the effect of P. gingivalis adhesion on subperiosteal implants. While in the introduction, the authors have explicitly discussed peri-implantitis (line 37). Is peri-implantitis also a side effect of infected subperiosteal implants? If not, authors are advised to include peri-implantitis as a complication of endosseous implants.
- When authors want to restrict this review to subperiosteal implants, it is essential to explain the infection of subperiosteal implants in the introduction without focusing more on peri-implantitis.
- Line 55 to 58, please include the reference of P. gingivalis since the review is focused on this bacteria and not other periodontal pathogens.
- Authors are suggested to include future directions of this effect surface topography-P. gingivalis adhesion together with conclusion. For example, how could surfacer topography be improved to decrease infection and promote the survival of implant materials?
- Check references 3, 10, 15, and 16,52, and make them similar to other references.
- Line 68; once it is in contact, line 201; have shown
Author Response
We have added few informations based on your suggestion. For further details please see attached files.

Round 2
Reviewer 1 Report
THE AUTHOR ADDRESSED ALL COMMENTS AND SUGGESTIONS. THEREFORE, I RECOMMEND THE ARTICLE FOR PUBLICATION.
Reviewer 2 Report
Thanks for revising the manuscript.